# Chemical Composition of Hazelnut Skin Food Waste and Protective Role against Advanced Glycation End-Products (AGEs) Damage in THP-1-Derived Macrophages

**DOI:** 10.3390/molecules28062680

**Published:** 2023-03-16

**Authors:** Ludovica Spagnuolo, Susanna Della Posta, Chiara Fanali, Laura Dugo, Laura De Gara

**Affiliations:** Department of Science and Technology for Sustainable Development and One Health, University Campus Bio-Medico of Rome, Via Alvaro del Portillo 21, 00128 Roma, Italy

**Keywords:** advanced glycation end-products (AGEs), circular bioeconomy, food waste, hazelnut skin, inflammation, polyphenols, reactive oxygen species (ROS)

## Abstract

Glycation and the accumulation of advanced glycation end-products (AGEs) are known to occur during aging, diabetes and neurodegenerative diseases. Increased glucose or methylglyoxal (MGO) levels in the blood of diabetic patients result in increased AGEs. A diet rich in bioactive food compounds, like polyphenols, has a protective effect. The aim of this work is to evaluate the capacity of hazelnut skin polyphenolic extract to protect THP-1-macrophages from damage induced by AGEs. The main polyphenolic subclass was identified and quantified by means of HPLC/MS and the Folin–Ciocalteu method. AGEs derived from incubation of bovine serum albumin (BSA) and MGO were characterized by fluorescence. Cell viability measurement was performed to evaluate the cytotoxic effect of the polyphenolic extract in macrophages. Reactive oxygen species’ (ROS) production was assessed by the H2-DCF-DA assay, the inflammatory response by real-time PCR for gene expression, and the ELISA assay for protein quantification. We have shown that the polyphenolic extract protected cell viability from damage induced by AGEs. After treatment with AGEs, macrophages expressed high levels of pro-inflammatory cytokines and ROS, whereas in co-treatment with polyphenol extract there was a reduction in either case. Our study suggests that hazelnut skin polyphenol-rich extracts have positive effects and could be further investigated for nutraceutical applications.

## 1. Introduction

Advanced glycation end-products (AGEs) are a heterogeneous group of compounds derived from spontaneous non-enzymatic glycation (or Maillard reaction) between reducing sugars and proteins, nucleic acids or lipids [1]. AGEs have been shown to cross-link with intracellular or extracellular proteins altering their physiological properties and functions [2], accumulate in cells or tissues contributing to chronic diseases, diabetic complications, cardiovascular diseases and are also involved in the progression of neurodegenerative diseases [3] such as Alzheimer’s disease (AD) and Parkinson’s disease (PD) [4,5], and during the physiological aging process or in autoimmune disease [6,7,8].

During the Maillard reaction, Amadori products formed [9] are converted directly to AGEs and others are oxidized to α-di-carbonyl compounds such as 3-deoxyglucosone (3-DG), methylglyoxal (MGO) and glyoxal (GO), which covalently bind to long-lived proteins to form stable AGE compounds [10]. In addition to endogenous AGEs’ formation, these compounds are also derived from foods high in lipids and proteins and are called dietary AGEs (dAGEs) [11]. In addition, various processes such us thermal food processing, storage, frying and cooking can contribute to increased AGEs’ content [12,13].

The adverse effects of AGEs on cellular functions includes several mechanisms such as the production of reactive oxygen species (ROS), oxidation of nucleic acids or lipids and interaction with specific receptors for AGEs (RAGE) expressed in different cell types [14]. AGE–RAGE interaction activates some intracellular signaling, such as transcription of nuclear factor kappa B (NF-kB), resulting in the production of cytokines, chemokines and other pro-inflammatory molecules that induce inflammation, apoptosis and proliferation [15,16,17]. In addition, AGEs accumulate in atherosclerotic lesions contributing to endothelial dysfunction and up-regulating the expression of vascular cell adhesion molecule-1 (VCAM-1) or intercellular adhesion molecule-1 (ICAM-1) [18,19].

Several synthetic drugs have been used to reduce accumulation of AGEs: for example, metformin reduces blood glucose levels and thus MGO levels, a precursor of AGEs. Aminoguanidine (AMG) acting as scavenger of α-di-carbonyl groups and blocks the conversion of Amadori products into AGEs. However, these drugs cause side effects that limit their application [20,21].

To date, many studies have suggested that natural compounds can inhibit formation of AGEs and reduce the harmful consequences of glycation [22,23,24,25]. Phenolic compounds are secondary metabolites of plants with good antioxidant activities that can neutralize undesired ROS and reactive nitrogen species (RNS) produced during metabolic processes in the body [26]. These natural compounds show a broad range of biological activities such as anti-inflammatory and anti-apoptotic activities, and inhibition of enzymes like α-amylase and glucosidase [27,28,29]. Different classes of polyphenolic compounds share the structural features of an aromatic ring and at least one hydroxyl group; they are classified by their chemical structures into flavonoids, phenolic acids, stilbenes and lignans [30,31].

Our work focuses on evaluating the protective effect of hazelnut skin polyphenols extract (HSE) on AGE-dependent damage in mammalian cell cultures such as the THP-1-macrophage cell line. Monocyte-macrophages have been selected as a cellular model because they perform important immunological functions and contribute to the maintenance of homeostasis and tissue repair [32].

The anti-glycation effect of hazelnut skin polyphenols has been previously reported in vitro in a chemical assay [33], thus suggesting a putative role of hazelnut skin extracts as natural drugs able to prevent the harmful glycation effects in vivo. Several works have shown that nut by-products are rich sources of natural phenolic compounds with bioactive potential [34,35]. Hazelnut (*Corylus avellana* L.) belongs to the Betulaceae family and is one of the most popular tree nuts consumed worldwide due to its nutrients, fat-soluble bioactive components, and phenols/phytochemicals [36]. Among hazelnut by-products, hazelnut skin has been previously characterized for its (poly)phenolic profile that provides the basis to investigate its potential health effects [37].

Therefore, the recovery of active compounds from food waste residues can be an interesting strategy of a circular bioeconomy, because in addition to being a natural and safe source of polyphenols, they are an inexhaustible, low cost, and sustainable resource [38].

## 2. Results

### 2.1. HPLC-PDA/ESI-MS Qualitative–Quantitative Analysis of Phenolic Compounds in Total Extracts of Hazelnut Skin

Since the level of phenolic compounds in plant tissues, apart being genetically determined, is strongly influenced by the pedo-climatic and environmental conditions in which the plant is grown, a preliminary qualitative–quantitative analysis of the phenolic molecules present in hazelnut skin was performed. The HPLC-PDA/ESI-MS method, previously described, was applied for phenolic compound determination in hazelnut skin. Qualitative analysis was performed considering the retention time, UV and MS spectra, use of standard compounds and data available in literature. Seventeen phenolic compounds have been identified in the extract. Twelve flavan-3-ols and two organic acids were detected at λ = 280 nm while four flavonols and one dihydrochalcone were detected at λ = 360 nm (Figure 1). For all identified compounds, retention time and their mass-to-charge ratio (*m*/*z*) are summarized in Table 1. Gallic acid, protocatechuic acid, procyanidin B2 dimer, (−) epicatechin, epigallocatechin-gallate, myricetin rhamnoside, quercetin-3-rhamnoside, kampferol rhamnoside, myricetin, phloretin-2-O-glucoside, quercetin and kaempferol were identified based on retention time of standard molecules and the mass-to-charge ratio (*m*/*z*) of the molecular ion.

For procyanidin beta type dimer gallate, procyanidin C2 trimer, prodelphinidin B dimer and procyanidin dimers, molecular standards were not available and only mass-to-charge ratio (*m*/*z*) of the molecular ion was considered. (+)-Catechin, and (−)-epicatechin, having the same mass-to charge ratio (*m*/*z*), were identified using purified (−) epicatechin as standard molecules and comparing the retention times. All the detected phenolic compounds were previously identified in hazelnut skin [33,37]. Calibration curves with external standard were constructed for each available standard molecule and linearity concentration range was between 1 and 100 mg/L for each curve. Phenolic compounds were quantified using the calibration curve of their standard molecule, if available, while procyanidins and prodelphinidins were quantified using the calibration curve of procyanidin B2 dimer, and (+) catechin using the calibration curve of (−) epicatechin. A total phenolic compound concentration of 445 mg/100 g was determined. As previously reported by Del Rio et al. [37], flavan-3-ols represent the main class of phenolic compounds in hazelnut skin. Procyanidin dimers resulted in being the two compounds present in high quantity with a concentration of 100 and 93 mg/100 g, respectively, followed by (+) catechin with a concentration of 62 mg/100 g. Among detected flavonols, quercetin-3-rhamnoside showed the highest quantity, with a concentration of 40 mg/100 g, confirming data reported in the literature [37]. The concentration of each identified phenolic compounds is reported in Table 2.

### 2.2. Total Phenolic Content (TPC)

Quantification of the total phenolic content (TPC) in food or biological samples is based on the reaction of phenolic compounds with a colorimetric reagent which allows measurement in the visible spectrum. This approach is considered to give an approximation of the real polyphenol content. Our results indicate that polyphenols in the hazelnut skin represent about 100 mg GAE/g, (10 g of polyphenol/100 g of hazelnut skin).

### 2.3. AGEs’ Quantification

Di-carbonyl compounds such as glycolaldehyde, GO, 3-DG and MGO, which are formed as intermediates during the glycation reaction, are more reactive and act as key components of carbonyl stress. In particular, MGO is an important precursor of AGEs that targets functional residues in proteins [39,40]. Several methods such as spectroscopy measurement or chromatography are available to determine parameters that are indicators of AGEs [41,42].

In our work, we have used the BSA–MGO model system for formation of AGEs, as mentioned in the materials and methods section. The presence of total AGEs in the sample was characterized by a fluorescence assay: the BSA–MGO sample showed a significant increase in specific AGE relative fluorescent units at λex 365 nm/λem 440 nm (Figure 2a). On the contrary, non-glycated BSA showed maximum fluorescent at λex 280 nm/λem 350 nm, as expected (Figure 2b).

### 2.4. Protective Role of HSE on Cell Viability Affected by AGEs

Glycation inhibitors derived from natural compounds are good candidates for the development of new therapies against diabetes and its complications and other pathological conditions related to AGE accumulation [43]. To assess whether the administration of the HSE could induce toxicity in a biological system, we performed cytotoxicity analysis. Macrophages were treated for 1 h with different concentrations of HSE and then with BSA–MGO at 300 µg/mL for 24 h. The MTT assay showed that HSE failed to display toxicity in macrophages up to a concentration of 400 µg/mL gallic acid equivalents (GAE) and only the administration of 500 μg/mL GAE was toxic (Figure 3). This could explain why phenolic compounds lose their antioxidant capacity at high concentration and start to behave as prooxidants [44].

On the contrary, BSA–MGO treatment resulted in the reduction of cell viability, in a dose-dependent manner (Figure 4b), whereas administration of BSA alone did not significantly reduce macrophages viability until 450 µg/mL (Figure 4a). In Figure 4a,b control is represented by untreated macrophages.

In the co-treatment, HSE protects against the reduction in viability following BSA–MGO treatment (Figure 5). Macrophages treated with BSA–MGO show a reduction in cell viability, which increases following HSE treatment at low concentration. When using HSE at high concentrations (from 100 to 400 µg/mL GAE) there seems to be a slight increase in viability even if it was not significantly different when compared to low concentrations (50 µg/mL GAE). In this context, in order to evaluate ROS scavenging activity and the inhibition of inflammation induced by AGE (used at 300 µg/mL) the concentration chosen for HSE was 50 µg/mL GAE.

### 2.5. Reduction of ROS by HSE

AGEs determine an increase in oxidative stress derived through different mechanisms of action [45]. Polyphenols have the ability to scavenge reactive carbonyl compounds and to donate an electron or hydrogen atom to free radicals [46,47]. Therefore, to evaluate if HSEs exhibit protective effects against ROS, we have stimulated macrophages with AGEs and our results show that BSA–MGO leads to an increase in ROS production slightly higher than that observed in the control cell culture (Figure 6). Interestingly, treatment with HSE remarkably inhibits the ROS production increase. In addition, as shown in Figure 6, the HSE at 50 μg/mL reduced the ROS production induced by BSA–MGO (Mix).

### 2.6. Modulation of Inflammatory Gene Expression by HSE

Inflammation plays a crucial role in the human body’s defense against pathogens and other harmful stimuli. However, uncontrolled inflammation can trigger activated macrophages to secrete excessive inflammatory mediators, leading to damage of otherwise healthy tissue [48].

Here, we have demonstrated that BSA–MGO (our AGEs’ model system) leads to a slight but significant increase in the gene expression of TNF-α, a key cytokine involved in acute inflammation while co-treatment with BSA–MGO and HSE (Mix) showed a reduction (Figure 7a). BSA–MGO treatment showed no effect on IL-1β gene expression, another mediator of the inflammatory response (Figure 7b). Treatment of cells with lipopolysaccharides (LPS) served as positive control for pro-inflammatory cytokine gene expression (Figure 7a,b).

### 2.7. Reduction in Pro-Inflammatory Cytokines’ Secretion

Production of the pro-inflammatory cytokines TNF-α and IL-1β in THP1 cells were determined after 24 h treatment. Results showed that HSE attenuated macrophage inflammation caused by BSA–MGO stimulation for both TNF-α and IL-1β secreted protein levels after co-treatments of cells with BSA–MGO and phenolic extract (Mix) (Figure 8a,b).

## 3. Discussion

A well-balanced diet in both macro- and micro-nutrients is an important factor in preventing or reducing chronic and systemic inflammation and can promote health, whereas an unhealthy diet can have the opposite effect [49]. In recent years the interest in plant based-foods, rich in bioactive molecules, has greatly increased based on the correlation with their positive effects on human health [50,51]. AGEs are a group of highly reactive chemical species, and their accumulation contributes to hyperglycemia, metabolic burden, increase in ROS production and inflammation which, along with insufficient production of endogenous antioxidants, induces oxidative stress leading to the development of chronic diseases [52,53].

Plant-derived polyphenols have been shown to have glucose lowering activities and, in some cases, direct AGE inhibitory functions [54]. Recently, many studies have proposed the antiglycation activities by polyphenols based on different properties including their structures, antioxidant abilities, and metabolism in the body [55,56]. The anti-glycation functions, depend on the ability to transfer electron free radicals, and are based on the total number of hydroxyl groups in the phenolic nuclear structure that can form stable radical complexes; such ability has been reported for some phenolic molecules [57].

The polyphenolic composition of hazelnut skins by HPLC-PDA/ESI-MS is comparable with the analysis carried out by Del Rio [37] with some differences in terms of the relative number of compounds found. These differences could depend on both the sensitivity of the analytical method used and the treatment of the raw material. Our analysis revealed that this food waste represents a rich source of several polyphenolic compounds some of which have already demonstrated a capacity to prevent AGEs’ related damaging effects. For example, it has been demonstrated that quercetin, a natural flavonol, could effectively inhibit the formation of AGEs in a dose-dependent manner via trapping reactive di-carbonyl compounds [58]. Phloretin prevented the formation of AGEs through trapping MGO in human umbilical endothelial cells (HUVECs) and also reduced inflammatory mediators [59]. Protocatechuic acid (phenolic acid) was effective in inhibiting formation of AGEs (BSA-glucose model system) resulting in a concentration dependent decrease [60]. Gallic acid (GA) showed a protective role in preventing AGE-mediated cardiac complications, through the attenuation of RAGE expression and also by modulating inflammatory downstream signaling pathways [61].

These and other compounds are present in abundance in hazelnut skin extract and therefore, we believe that the inhibitory effect on AGEs’ formation, as previously demonstrated [33] and the positive effect on macrophages obtained in this study, can be attributed to the synergistic effect of the (poly)phenolic compounds present in the total extract.

Our data showed that HSE is able to counteract AGEs’ induced damage in a mammalian cell culture in vitro model. In our study, BSA–MGO (AGEs’ model system) lead to an increase in pro-inflammatory cytokines, such as TNF-α and IL-1β, in macrophages (M0). In particular AGEs determine an increase in IL-1β secretion comparable to LPS stimulation, indicating that these macromolecules can contributed to inflammation increments. HSE exerts anti-inflammatory activity, evident by the reduction in the expression of TNF-α (Figure 7a). Protein secretion induced by AGE of both TNF-α and IL-1β was significantly reduced after administration of HSE. We have also demonstrated the antioxidant capacity of HSE, which reduced ROS production in macrophages stimulated by AGEs. The administration of HSE in macrophages alone or in combination with AGEs reduce significantly the amount of ROS, directly proportionally to fluorescent measures. On the contrary, AGEs stimulate ROS production more than the control.

These results confirm that HSE have specific anti-glycation activities in both pro-inflammatory and pro-oxidant pathways activated by AGEs and probably also correlate with the phenolic structure as described above. However, we believe that the complex composition of HSE may potentiate the efficacy leading to a greater effect than using a pure compound.

The ability of HSE to prevent AGEs’ formation previously demonstrated [33], and the results reported here, strongly support a promising role for the use of HSE in all the diseases characterized by excessive production of AGEs.

Our results support the possibility of using this food waste in the context of a circular bioeconomy as an interesting source of anti-AGEs and health-promoting compounds.

We are willing to further investigate whether HSE is able to reduce the inflammatory response caused by AGEs, by altering the AGE–RAGE signaling pathway leading to the activation of NF-κB, a ubiquitous transcription factor involved in several diseases. This will be the next step towards a better understanding of the ability of HSE to decrease the AGEs’ dependent damage occurring in mammal cells [62].

## 4. Materials and Methods

### 4.1. Preparation of HSE

The HSE was obtained following the procedure reported by Del Rio et al. [37] with some modifications. Two sequential extractions were applied: an amount of 0.5 g of hazelnut skins was added to 5 mL of 1% (*v*/*v*) aqueous formic acid solution in a 15 mL centrifuge tube and extracted for 30 min in an ultrasonic bath (Elmasonic S30H, Elma Schmidbauer GmbH, Singen, Germany) at room temperature, with a frequency of 37 kHz, and a heating power of 200 W. Then, the sample was heated at 70 °C for 1 h and centrifuged for 10 min at 2151× *g*. The procedure was repeated two times, the supernatant was recovered, and a second extraction was performed on the sample remaining after centrifugation. A solution of 5 mL of methanol/H_2_O (75:25, *v*/*v*) was added and it was placed for 15 min in an ultrasonic bath (Elmasonic S30H, Elma Schmidbauer GmbH, Singen, Germany) and vortexed for 15 min. This procedure was repeated twice. After centrifugation, 10 min at 2151× *g*, the supernatant was recovered and added to the extract obtained with the first extraction step. The solvent of the total extract was evaporated under vacuum at 30 °C in a rotary evaporator (Eyela, Tokyo, Japan). The extract was then dissolved in 1 mL of MeOH and analyzed using HPLC–PDA/ESI–MS.

### 4.2. HPLC–PDA/ESI–MS Analysis of HSE

HSEs were analyzed using a Shimadzu Prominence LC-20A instrument (Shimadzu, Milan, Italy) equipped with two LC-20 AD XR pumps, SIL-10ADvp, CTO-20 AC column oven, and DGU-20 A3 degasser coupled to an SPD-M10Avp PDA detector and a mass spectrometer detector (LCMS-2010, Shimadzu, Tokyo, Japan) equipped with an electrospray (ESI) interface. Separation was performed using a Core Shell column (150 µm, 4.6 mm I.D., 2.7 µm d.p.) (Merck KGaA, Darmstadt, Germany), with the mobile phase composed of 1% aqueous formic acid (A) and acetonitrile (B), pumped at a flow rate of 1 mL/min. Phenolic compound separation was obtained by applying the following gradient: t = 00 0%B; t = 40 30%B; t = 41 100%B; t = 48 100%B; t = 49 0%B; t = 56 0%B. The injection volume was 2 µL. Data were acquired using a PDA in the range 210–400 nm and chromatograms were extracted at 360 nm and at 280 nm. MS-chromatograms were acquired in negative ionization mode, using the following parameters: nebulizing gas flow rate (N2): 1.5 mL min^−1^; event time: 1 s; mass spectral range: *m*/*z* 100–800; scan speed: 1000 amu/s; detector voltage: 1.5 kV; interface temperature: 250 °C; CDL temperature: 300 °C; heat block temperature: 300 °C; interface voltage: −3.50 kV; Q-array voltage: 0.0 V; Q-array RF: 150.0 V.

### 4.3. Determination of Total Phenolic Content

The total extract of hazelnut skin was dried as described above and resuspended in 5 mL of methanol/H_2_O (50:50; *v*/*v*) solution, filtered with 0.22 µm filter under sterile conditions for use in cell cultures and subsequently analyzed for the total polyphenols content through a chemical reduction, measured by the Folin–Ciocalteau method [63]. Briefly, an aliquot of 20 µL of extract or standard compound was mixed with 100 µL of Folin reagent in 1580 µL of methanol/H_2_O (50:50; *v*/*v*) solution, followed by incubation for 8 min. Then, 300 µL of Na_2_CO_3_ 0.2 g/mL solution was added. The absorbance was measured after incubation at room temperature for 2 h, in the dark using a microplate reader (Infinite 200 Pro, Tecan, Männedorf, Switzerland). The total phenolic content was determined from a standard curve using gallic acid (0−2000 µg/mL) as a standard and expressed as milligrams of gallic acid equivalents per grams of hazelnut fresh weight (mg GAE/g). All chemicals and reagents were purchased from Sigma (Sigma-Aldrich, Milan, Italy).

### 4.4. Cell Culture and Differentiation

The THP-1 cell line (passages 6–20, ATCC: TIB-202) was cultured in RPMI 1640 medium (Corning, NY, USA) supplemented with 100 U/mL penicillin, 100 µg/mL streptomycin (Corning, NY, USA) 10 mM HEPES (Dominique Dutscher, Bernolsheim, France), 2 mM L-glutamine (Corning, NY, USA), and 10% (*v*/*v*) heat-inactivated fetal bovine serum, FBS (Corning, NY, USA) at 37 °C in a humidified atmosphere containing 5% (*v*/*v*) CO_2_. Routinely, THP-1 cells were cultured in T75 flasks and sub-cultured every three to four days at a concentration of 4 × 10^5^ to 1 × 10^6^ cells/mL. THP-1 cells can be differentiated into macrophage-like cells using 100 ng/mL phorbol 12-myristate 13-acetate PMA (Sigma-Aldrich, Milan, Italy) for 72 h. During this time, cell attach to the bottom of the cell culture plates and develop macrophage-like morphology. After macrophage differentiation, cells rest for another 24 h in the culture medium without PMA to obtain the resting state of macrophages (M0).

### 4.5. Preparation of Glycated BSA

AGE-BSA was prepared by reacting bovine serum albumin (BSA, Sigma-Aldrich, Milan, Italy) with MGO (Sigma-Aldrich, Milan, Italy) according to the method described by Starowicz et al. with some modifications [64]. The method and analysis were validated in our previous work [33]. Briefly, BSA (4 mg/mL) was dissolved in phosphate buffered saline 1×, pH 7.4 (PBS, Corning, NY, USA) with the addition of 1% pen/strep (to prevent microbe development) and filtered with a 0.22 µm filter, under sterile conditions. BSA with or without addition of 20 mM MGO (BSA glycated and BSA-non-glycated, respectively) were incubated for 168 h at 37 °C, in the dark. Glycation was confirmed by fluorescence of the BSA–MGO model system (AGEs) or BSA measured after incubation using a microplate reader (Infinite 200 Pro, Tecan, Männedorf, Switzerland) at excitation/emission wavelengths 365/440 nm. Additionally, changes in intrinsic protein fluorescence were detected at excitation/emission wavelengths of 280/350 nm. Quantification of the BSA–MGO or BSA samples was performed by the BCA assay (Merck, Darmstadt, Germany) according to the protocol.

### 4.6. Cell Viability

Differentiated THP-1 macrophages (25,000 cells/well in 96-well plates) were exposed to different concentrations of HSE (25–50–100–200–400–500 µg/mL GAE) and with BSA or BSA–MGO (100–150–300–450–600 µg/mL). Untreated cells represented the control group. Cells were treated with or without HSE for 1 h prior to BSA–MGO (300 µg/mL) treatment. Cytotoxicity was determined after 24 h of incubation by the MTT assay: cell culture medium was discarded, and each well was washed with 200 μL PBS. MTT solution 0.5 mg/mL (Sigma-Aldrich, Milan, Italy), was added to cells (100 µL in each well) and the plate was incubated at 37 °C + 5% CO_2_ for 3 h. Then, MTT solution was removed, and 150 μL/well of dimethyl sulfoxide (DMSO, Sigma-Aldrich, Milan, Italy) was added to each well to dissolve the formazan crystals. Optical density (OD) was measured at 570 nm using a multifunctional microplate reader (InfiniteM 200 Pro, TECAN, Männedorf, Switzerland). Cell viability was expressed as % of control (untreated macrophages).

### 4.7. ROS Measurement

For measuring intracellular ROS, macrophages were used at density of 5 × 10^4^ cells/well in 96-well black plates. Cells were loaded with 100 µL/well of H2DCF-DA (Merck, Darmstadt, Germany) diluted to 20 µM in PBS and incubated for 10 min at 37 °C. Then the solution with probe was removed and replaced by 100 µL/well basal medium. Basic fluorescence intensity was measured at 495 nm excitation and 525 nm emission. Medium was removed and treatments were applied. After 60-min, fluorescence intensity was measured as described above [65].

### 4.8. Quantitative Real-Time PCR

Total RNA was isolated from cells by a column-based method, Monarch Total RNA Miniprep Kit (New England BioLabs, Frankfurt am Main, Germany) according to manufacturer’s instructions including DNAse I-treatment. Concentration and quality of isolated RNA was spectrophotometrically assessed by NanoDrop (Thermo Fisher Scientific Inc., Waltham, MA, USA). Ready-to-use kit (Thermo Fisher Scientific) was used to reverse transcribe 0.5 µg RNA, according to manufacturer’s protocol. Quantitative real-time PCR was performed using the SYBR green master mix (Applied Biosystems, Waltham, MA, USA). Primer sets for TNF-α (NM_000594.4; Fw: 5′-CTCCTCACCCACACCATCAGCCGCA-3′; Rv: 5′-ATAGATGGGCTCATACCAGGGCTTG-3′) and IL-1β (NM_000576.3; Fw: 5′-CTCGCCAGTGAAATGATGGCT-3′; Rv: 5′-TGGTGGTCGGAGATTCGTAGC-3′), were used. Actin-β (NM_001101.5; Fw: 5′-GGGAAATCGTGCGTGACATT-3′; Rv: 5′-TCGTAGATGGGCACAGTGT-3′) was used as a housekeeping gene to normalize data. All reactions were performed in triplicates. ΔΔCt method was used for data analysis. Values of genes of interest were first substracted from the values of actin-β (ΔCt). N-fold change of gene expression was then calculated as 2^−(ΔCt treated-ΔCt untreated)^.

### 4.9. Cytokine Quantification

The levels of human TNF-α and IL-1β were measured in the harvested supernatants by solid-phase sandwich Enzyme-Linked Immunosorbent Assay (ELISA kit; Thermo Fisher Scientific, Waltham, MA, USA). THP-1 monocytes were seeded into six-well plates (1 × 10^6^ cell/well) and differentiated into macrophages-M0. Cells were treated with 50 µg/mL HSE or 300 µg/mL BSA–MGO or in combination. After 24 h cell supernatant for each sample was collected and analyzed according to the manufacturer’s recommendation. Optical density (OD) was measured using the microplate reader (Infinite M 200 Pro, TECAN, Männedorf, Switzerland) at 450 nm.

### 4.10. Statistical Analysis

The graphics and all statistical analyses were performed using GraphPad Prism version 5.0. The data were expressed as mean ± standard deviation of two/three independent experiments, with at least three technical replicates in each experiment. *p*-value, * *p* < 0.05, ** *p* < 0.01, *** *p* < 0.001 were considered statistically significant and were calculated using one-way ANOVA and Tukey as post-test or *t*-test of multiple or two comparisons, respectively. Different letters indicate a significative difference among samples.

## 5. Conclusions

Glycation or Maillard reaction is a complex of spontaneous, non-enzymatic reactions that form compounds described as AGEs. Exogenous as well as endogenous AGEs interact with specific receptors resulting in the activation of a series of signaling pathways implicated in inflammation and progression of chronic and degenerative diseases [66]. Discovery of new potential anti-glycation agents, natural or synthetic, represents an effective approach to control the development and prevention of disease linked to AGEs’ accumulation. Several evidence have shown that diets rich in plant food are protective against a wide range of health conditions. Indeed, intake of flavonoid-rich foods has been shown to be very beneficial to human health [52].

Data obtained show that polyphenols in hazelnut skin have a protective effect in macrophages following AGE stimulation and could potentially pair with, or replace synthetic drugs. Because of the contribution of AGEs to the pathogenesis of several chronic diseases, the use of natural bioactive molecules could represent an interesting new therapeutic strategy with positive effects on human health.

## Figures and Tables

**Figure 1 molecules-28-02680-f001:**
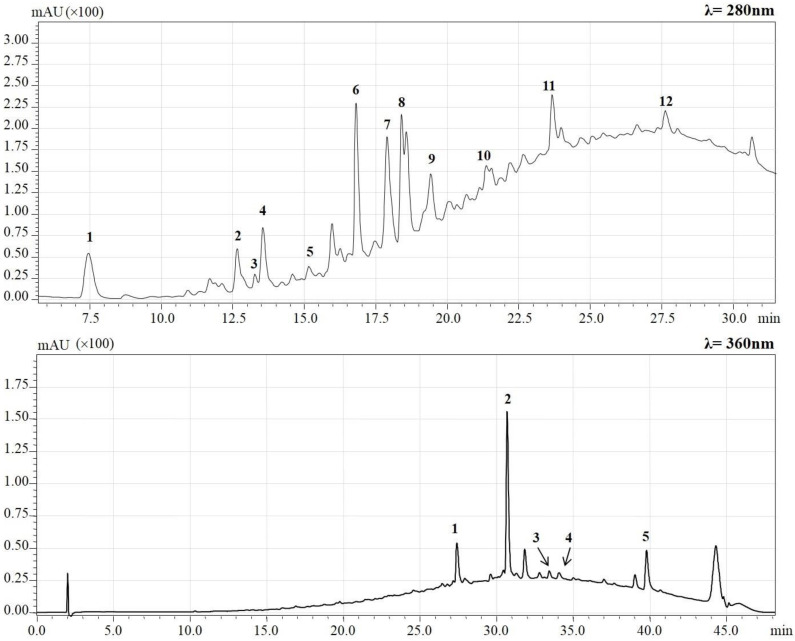
HPLC-PDA chromatograms of hazelnut skin phenolic compound profile, measured at λ = 280 nm and λ = 360 nm. Peak numbers refer phenolic compounds reported in detail in Table 1.

**Figure 2 molecules-28-02680-f002:**
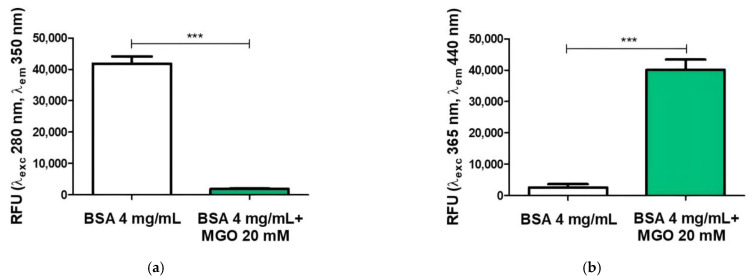
Fluorescence measurement of AGEs after 168 h of incubation. Data represent relative fluorescence units (RFU) at (**a**) λex 365 nm/λem 440 nm and (**b**) λex 280 nm/λem 350 nm for BSA non-glycated (BSA) and BSA glycated (BSA + MGO). The analysis of variance was carried out using *t*-test analysis; *** *p* < 0.001 value was considered significant (BSA vs. BSA-MGO). All values are expressed as M ± SD (n = 3).

**Figure 3 molecules-28-02680-f003:**
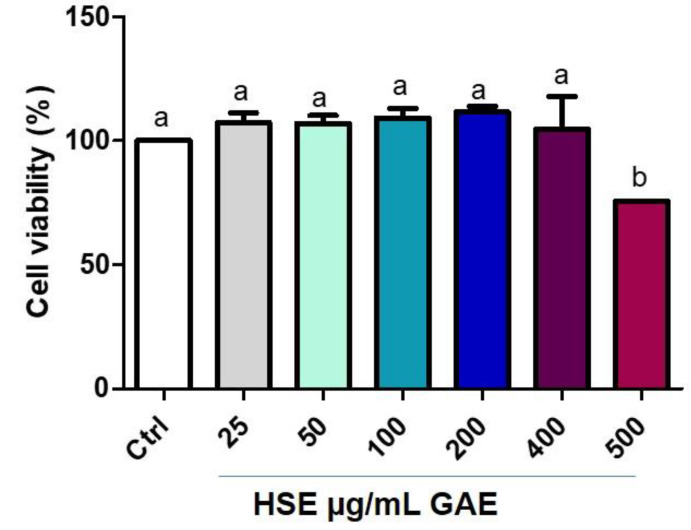
Cell viability after HSE treatment on THP-1 macrophages (M0). The MTT assay was performed at 24 h post-treatment. Cell viability was expressed as % of control (non-treated). All data represent the means of at least three replicates ± standard deviation. The analysis of variance was carried out using one-way ANOVA followed by the Tukey’s multiple comparison test: letters denote significant differences among samples.

**Figure 4 molecules-28-02680-f004:**
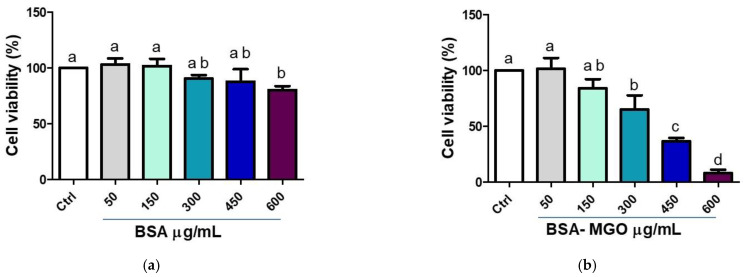
Cell viability after (**a**) BSA or (**b**) BSA–MGO treatment on THP-1 macrophages (M0). MTT assay was performed at 24 h post-treatment. Absorbance was read at 570 nm and cell viability was expressed as % of control (untreated). All data represent the means of at least three replicates ± standard deviation. The analysis of variance was carried out using one-way ANOVA followed by the Tukey’s multiple comparison test: letters denote significant differences among samples (Ctrl vs. 300 µg/mL BSA *p* = ns; Ctrl vs. 300 µg/mL BSA–MGO *p* < 0.01).

**Figure 5 molecules-28-02680-f005:**
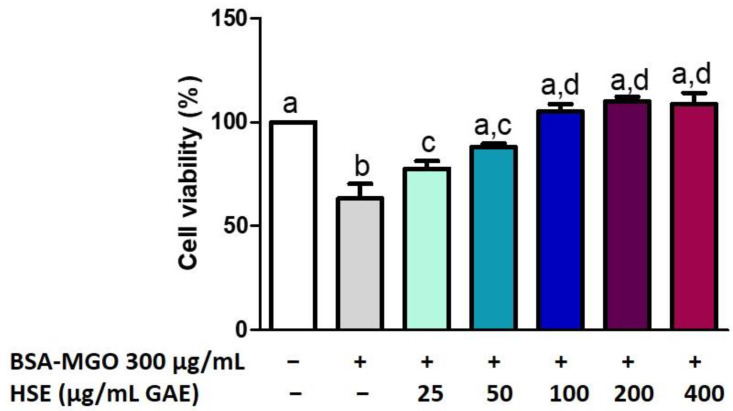
Cell viability after HSE co-treatment with BSA–MGO on THP-1 macrophages (M0). MTT assay was performed at 24 h post-treatment. Cell viability was expressed as % of control (non-treated). All data represent the means of at least three replicates ± standard deviation. The analysis of variance was carried out using the one-way ANOVA test followed by the Tukey’s multiple comparison test: letters denote significant differences among samples.

**Figure 6 molecules-28-02680-f006:**
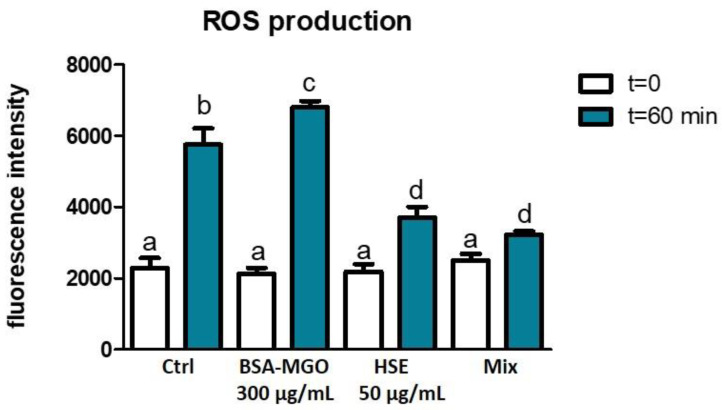
ROS production in macrophages. THP-1 macrophages M0 were treated with BSA-MGO, HSE or in combination (MIX) for up to 60 min. Production of intracellular ROS was determined using fluorescent probe H2-DCFDA and measurement of fluorescence intensity. Basic measurement (t = 0) represents fluorescence intensity after loading with H2-DCFDA, without addition of any treatment. All data represent the means of at least three replicates ± standard deviation. The analysis of variance was carried out using the one-way ANOVA followed by the Tukey’s multiple comparison test: letters denote significant differences among samples (*p* < 0.001 BSA–MGO vs. MIX, t = 60 min).

**Figure 7 molecules-28-02680-f007:**
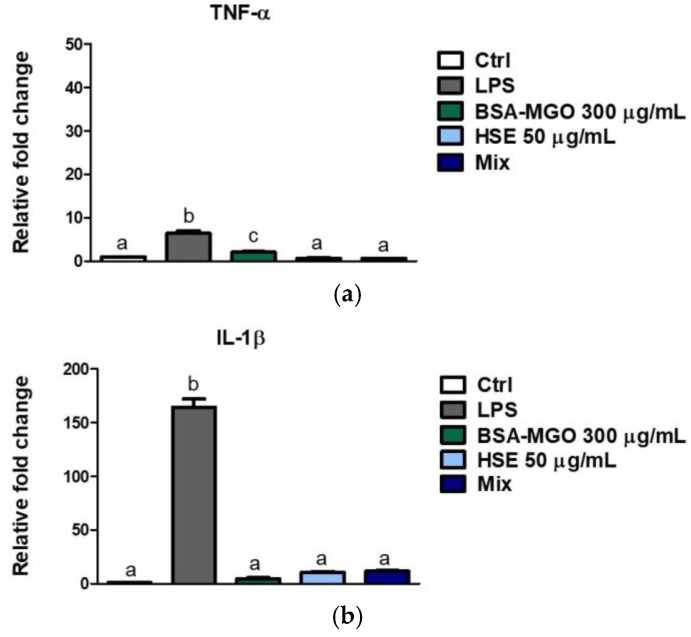
Expression of TNF-α and IL-1β. THP-1 macrophages (M0) were treated with BSA-MGO, HSE or a combination (MIX). Expression of (**a**) TNF-α and (**b**) IL-1β genes were quantified using qPCR after 24 h. Data were normalized to untreated control cells. All data represent the means ± SD of relative mRNA expression of two independent experiments. The analysis of variance was carried out using one-way ANOVA followed by Tukey’s multiple comparison test: letters denote significant differences among samples. *p* < 0.01 BSA–MGO vs. MIX for TNF-α; *p* = ns BSA–MGO vs. MIX for IL-1 β).

**Figure 8 molecules-28-02680-f008:**
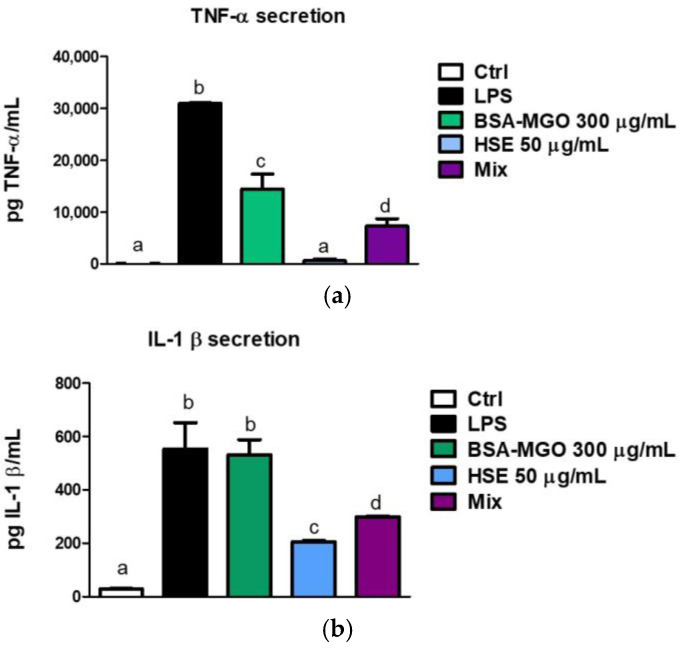
Quantification of pro-inflammatory cytokines TNF-α and IL-1β. THP-1 macrophages (M0) were treated with BSA-MGO, HSE or in combination (MIX). Levels of (**a**) TNF-α and (**b**) IL-1β secreted proteins were determined by ELISA assay, after 24 h. Data were expressed as pg standard/mL. All data represent the means ± SD of two independent experiments. The analysis of variance was carried out using one-way ANOVA followed by the Tukey’s multiple comparison test: letters denote significant differences among samples. *p* < 0.01 BSA–MGO vs. MIX for TNF-α; *p* < 0.05 BSA–MGO vs. MIX for IL-1 β).

**Table 1 molecules-28-02680-t001:** HPLC-PDA/ESI-MS identification of phenolic compounds in hazelnut skin extract.

N	Phenolic Compound	t_R_	*m*/*z* [M−H]^−^
Detected at λ = 280 nm			
1	Gallic acid	7.4	169
2	Protocatechuic acid	12.6	153
3	Prodelphinidin B dimer	13.3	593
4	Procyanidin C2 trimer	13.5	865
5	Prodelphinidin B dimer	15.1	593
6	Procyanidin dimer	16.8	577
7	Procyanidin dimer	17.9	577
8	(+) Catechin	18.4	289
9	Procyanidin B2 dimer	19.4	577
10	(−) Epicatechin	21.3	289
11	Procyanidin beta type dimer gallate	23.7	729
12	Epicatechin gallate	27.6	441
Detected at λ = 360 nm			
1	Myricetin rhamnoside	27.3	463
2	Quercetin 3-rhamnoside	30.6	447
3	Kampferol rhamnoside	33.4	431
4	Phloretin-2-o-glucoside	34.9	435
5	Quercetin	39.7	301

**Table 2 molecules-28-02680-t002:** Phenolic compound content in hazelnut skin extract.

Phenolic Compound	Concentration mg 100 g^−1^ ± SD
Gallic acid	15.683 ± 0.159
Protocatechuic acid	13.908 ± 1.440
Prodelphinidin B dimer	13.477 ± 0.000
Procyanidin C2 trimer	4.197 ± 0.058
Prodelphinidin B dimer	21.189 ± 0.400
Procyanidin dimer	100.480 ± 3.207
Procyanidin dimer	92.652 ± 1.115
(+) Catechin	62.137 ± 4.732
Procyanidin B2 dimer	15.552 ± 0.515
(−) Epicatechin	7.173 ± 0.000
Procyanidin beta type dimer gallate	16.803 ± 0.135
Epicatechin gallate	0.828 ± 0.099
Myricetin rhamnoside	16.465 ± 0.667
Quercetin-3-o-rhamnoside	39.623 ± 1.176
Kampferol rhamnoside	2.504 ± 0.048
Phloretin-2-o-glucoside	12.595 ± 0.568
Quercetin	10.658 ± 0.659
Total phenolic compounds	445.923 ± 7.312

## Data Availability

Not applicable.

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
