# Peer review of "Chemical Composition of Hazelnut Skin Food Waste and Protective Role against Advanced Glycation End-Products (AGEs) Damage in THP-1-Derived Macrophages"

_molecules, 2023, doi:10.3390/molecules28062680_

Round 1

Reviewer 1 Report

The article is interesting and generally well written, although editing is required to correct minor errors in language, plurality and word usage. The methodology is contemporary and described in detail. All abbreviations/acronyms need to be defined when first used, and in figures and tables. There is some repetition in that the impacts of AGEs are stated in multiple paragraphs in the Introduction and in the Discussion.

Line 113 - why is the analysis described as "qualitative"?

Line 189 - viability was reduced significantly at 600 ug/mL of BSA.

Line 203 - change "contest" to "context".

Line 220 - change "Figure 5" to "Figure 6".

Lines 241-242 - LPS needs to be defined; change "Figure 5 a, b" to "Figure 7 a, b".

Author Response

Reviewer 1

We have revised all text and correct minor errors as suggested by reviewer

 Comment 1: Why is the analysis described as "qualitative"?

A: Through HPLC-PDA/ESI-MS a quali-quantitative analysis was performed. As described in the paragraph 2.1 (line 119), the qualitative analysis was performed to identify phenolic compounds in hezelnut skin while quantitative analysis was performed to obtain an estimate of the quantity of each phenolic compound by calibration curves with available standard molecules (line 141)

Comment 2: viability was reduced significantly at 600 ug/mL of BSA.

A: we have modified the text in line 202

Comment 3: change "contest" to "context".

A: we have changed the word; line 218

Comment 4: change "Figure 5" to "Figure 6".

A: we have changed the number; line 236

Comment 5: LPS needs to be defined; change "Figure 5 a, b" to "Figure 7 a, b".

A: Done. Line 258 and 259

Reviewer 2 Report

This manuscript primarily evaluates the capacity of hazelnut skin polyphenolic extract to protect THP-1-macrophages from AGEs-induced damage. The main polyphenolic subclass was identified and quantified by means of HPLC/MS and Folin–Ciocalteu method. AGEs derived from the incubation of bovine serum albumin (BSA) and MGO were characterized by fluorescence. The final study showed that polyphenolic extract protected cell viability from AGEs-induced damage. After treatment with AGEs, macrophages express high levels of pro-inflammatory cytokines and ROS, whereas in co-treatment with polyphenol extract there is a reduction in either case.

It is an interesting topic regarding the positive effect of hazelnut skin polyphenols-rich extracts on cell damage triggered by AGEs. However, there still have some issues which need to be cleared.

1.     The title is too long and should be appropriately trimmed.

2.     The introduction should be compress, however the polyphenols inhibit AGEs mechanism should be analyzed (Insights into oat polyphenols against advanced glycation end products mechanism by spectroscopy and molecular interaction. Food Bioscience. 43(2021), 101313.).

3.     AGEs can damage cells and promote the expression of high levels of inflammatory factors and ROS, which occurs especially during aging, diabetes, and neurodegenerative diseases (Dietary polyphenols: regulate the advanced glycation end products (AGEs)-RAGE axis and the microbiota-gut-brain axis to prevent neurodegenerative diseases. Critical Reviews in Food Science and Nutrition. Doi: 10.1080/10408398.2022.2076064).

4.     “Materials and methods” should come before “results” in a manuscript.

5.     Why AGEs apply BSA and MGO for incubation?

6.     Results “2.1. HPLC-PDA/ESI-MS Quali-Quantitative Analysis of Phenolic Compounds in total Extracts of Hazelnut Skin” in the meanwhile the CML should be analyzed (Quantitative determination of Nepsilon-(carboxymethyl)lysine in sterilized milk by isotope dilution UPLC-MS/MS method without derivatization and ion pair reagents. Food chemistry. 385(2022): 132697.).

7.     “2.2. Total phenolic content(TPC)” should be used as a subheading of “2.1. HPLC-PDA/ESI-MS Quali-Quantitative Analysis of Phenolic Compounds in total Extracts of Hazelnut Skin”.

8.     In the control of Figure 2-a,b, only BSA was added?

9.     What do the results of b indicate? Quenching of fluorescence

10.  “2.5. Reduction of ROS by HSE” the ROS test method should cite this reference(Consumption of the fish oil high-fat diet uncouples obesity and mammary tumor growth through induction of reactive oxygen species in pro-tumor macrophages. Cancer Research, 2020, 80(12): 2564-2574.).

11.  The font of the figure should be larger, in addition, the scale value of the vertical coordinate in Figure 7(b) below can be adjusted, which will make the picture more beautiful.

12.  The first four paragraphs of the discussion are devoted to polyphenols and the results of others. In this section, the results of the experiment should be discussed and the prospect should be presented for the experimental results, therefore, the discussion section should be reorganized and rewritten.

The reference should be updated and the manuscript should be revised before it can bepublished.

Author Response

Reviewer 2

Comment 1: The title is too long and should be appropriately trimmed.

A: we have modified the title

 Comment 2: The introduction should be compress, however the polyphenols inhibit AGEs mechanism should be analyzed (Insights into oat polyphenols against advanced glycation end products mechanism by spectroscopy and molecular interaction. Food Bioscience. 43(2021), 101313.).

A: we have modified the introduction. We have already analyzed the inhibitory effect of hazelnut skin extract on AGEs formation in a previously published article (ref. 33, Spagnuolo et al, 2021), as reported in line 90. We have inserted the suggested citation at line 167

Comment 3: AGEs can damage cells and promote the expression of high levels of inflammatory factors and ROS, which occurs especially during aging, diabetes, and neurodegenerative diseases (Dietary polyphenols: regulate the advanced glycation end products (AGEs)-RAGE axis and the microbiota-gut-brain axis to prevent neurodegenerative diseases. Critical Reviews in Food Science and Nutrition. Doi: 10.1080/10408398.2022.2076064).

A: we have added the suggested references, line 38

Comment 4: “Materials and methods” should come before “results” in a manuscript.

A: we have consulted the “Molecules” website and used the Microsoft word template available in the instruction for author’s section. The official template indicates the following sequence: 1) introduction; 2) results; 3) discussion; 4) materials and methods; 5) conclusions

Comment 5: Why AGEs apply BSA and MGO for incubation?

A: MGO is a reactive precursor of AGE, in literature some authors have used MGO to develop a model of AGEs formation. The incubation with MGO is reported to last about seven days, whereas the incubation with glucose is longer. In our previously published work (ref. 33, Spagnuolo et al, 2021), we have characterized AGEs using different concentrations of MGO and different incubation time and have choosen this precursor to producing final AGE as the method has proven to work well and it is convenient in terms of time saving.

Comment 6: Results “2.1. HPLC-PDA/ESI-MS Quali-Quantitative Analysis of Phenolic Compounds in total Extracts of Hazelnut Skin” in the meanwhile the CML should be analyzed (Quantitative determination of Nepsilon-(carboxymethyl)lysine in sterilized milk by isotope dilution UPLC-MS/MS method without derivatization and ion pair reagents. Food chemistry. 385(2022): 132697.).

Thank you for this suggestion, it would be quite interesting to carry out this measurement. Unfortunately, the analytic instrumentation currently available to our research group do not allow the fragmentation of complex analytes and to carry out this kind of targeted analysis. We hope we’ll be able to acquire more powerful instrumentation in the next future as a triple quadrupole MS.

We add the suggested article in line 167.

Comment 7: “2.2. Total phenolic content (TPC)” should be used as a subheading of “2.1. HPLC-PDA/ESI-MS Quali-Quantitative Analysis of Phenolic Compounds in total Extracts of Hazelnut Skin”

A: We have considered two sections because HPLC and Folin represent two different methods. The second method is used to obtain an approximation of phenolic content, but it is less specific.

Comment 8: In the control of Figure 2-a,b, only BSA was added?

A: The control “ctrl” represents macrophages without any treatment. We added the specification in line 202-203 and in line 453.

Comment 9: What do the results of b indicate? Quenching of fluorescence

A: In figure 4b as well as in figure 4a results were obtained by absorbance at 570 nm (optical density) and then expressed as % of control (untreated cells), as described in materials and method (line 460). We have added the absorbance value at line 208.

Comment 10: 2.5. Reduction of ROS by HSE” the ROS test method should cite this reference (Consumption of the fish oil high-fat diet uncouples obesity and mammary tumor growth through induction of reactive oxygen species in pro-tumor macrophages. Cancer Research, 2020, 80(12): 2564-2574.).

A: we added the reference as suggested at line 232

Comment 11: The font of the figure should be larger, in addition, the scale value of the vertical coordinate in Figure 7(b) below can be adjusted, which will make the picture more beautiful.

A: Done. We have modified all figure as suggested

Comment 12: The first four paragraphs of the discussion are devoted to polyphenols and the results of others. In this section, the results of the experiment should be discussed, and the prospect should be presented for the experimental results, therefore, the discussion section should be reorganized and rewritten.

A: thank you for this suggestion. We have now modified and rewritten the discussion

Round 2

Reviewer 2 Report

The authors of this manuscript has responsed the reviewer's comments point by point. It can be accepted in current revision.